

# Comparison of shallow-water seston among biogenic habitats on tidal flats

Jennifer L. Ruesink[1], Cinde R. Donoghue[2], Micah J. Horwith[2], Alexander T. Lowe[1] and Alan C. Trimble[1]

[1] Department of Biology, University of Washington, Seattle, WA, United States of America
[2] Washington Department of Natural Resources, Olympia, WA, United States of America

## ABSTRACT

Aquatic structure-formers have the potential to establish mosaics of seston in shallow water if they modify the relative amounts of deposition (or filtration) and resuspension of particles. By sampling surface water adjacent to Lagrangian drifters traveling 0.1 to 2 m above the bottom, we tested the modification of seston in water masses flowing over two biogenic marine species (native eelgrass, *Zostera marina*; introduced oysters, *Crassostrea gigas*) in comparison to unstructured tidal flats. Water properties were examined at five intertidal sites in Washington State, USA, each with 27 drifts (three drifts at different stages of the tidal cycle in each of three patches of three habitat types; drift distance 116 m (109SD), duration 24 min (15SD)). At the initiation of each drift, habitat differences in water properties were already apparent: chlorophyll-*a* and total suspended solid (TSS) concentrations were greater in structured habitats than bare, and TSS was also inversely related to water depth. Water flowed more slowly across eelgrass than other habitat types. As water flowed across each habitat type, TSS generally increased, especially in shallow water, but without habitat differences; chlorophyll-*a* in these surface-water samples showed no consistent change during drifts. At higher TSS concentrations, quality in terms of organic content declined, and this relationship was not habitat-specific. However, quality in terms of chlorophyll-*a* concentration increased with TSS, as well as being greater in water over eelgrass than over other habitat types. These results support widespread mobilization of seston in shallow water ebbing or flooding across Washington State's tidal flats, especially as water passes into patches of biogenic species.

## INTRODUCTION

Intertidal organisms, in addition to spending time in both water and air, experience short-term variation because coastal water fluctuates in such properties as dissolved gas concentrations (*Duarte et al., 2013*) and particle loads (*Ralph et al., 2007*). The amount and composition of suspended particulate matter, termed seston, has important implications for the productivity of macrophytes via light limitation (*DeBoer, 2007*) and for benthic suspension feeders via food resources (e.g., *Kang et al., 2003*). Quantifying how biogenic species influence the mobilization and removal of particles improves understanding of the feedbacks governing local heterogeneity in water properties (*Widdows et al., 2008*; *DeBoer,*

Corresponding author
Jennifer L. Ruesink,
ruesink@u.washington.edu

*2007*). In this study, we track water properties across tidal flats and compare unstructured habitat to two dominant structure-forming species: eelgrass (*Zostera marina*) and oysters (*Crassostrea gigas*).

These two species are expected to modify water properties in different ways. As suspension-feeders, oysters remove particles, and water passing across oyster beds typically declines in chlorophyll concentration (*Grizzle et al., 2006*; *Grizzle, Greene & Coen, 2008*; *Grangere et al., 2010*; *Plutchak et al., 2010*; *Wheat & Ruesink, 2013*. Away from the footprint of oysters, mixing and compensatory phytoplankton dynamics obscure this trophic effect (*Dame & Libes, 1993*; *Plutchak et al., 2010*). Seagrass influences seston predominantly as a side effect of altering water motion, since more rapid flow or turbulence can lift and transport larger, denser particles (*Widdows et al., 2008*; *Wilkie et al., 2012*). Empirically, seagrass beds buffer against water motion and accumulate fine particles (*Fonseca et al., 1982*; *Kenworthy, Zieman & Thayer, 1982*). However, effects on flow velocity and turbulence within and around structure are sensitive to a variety of parameters that naturally vary in the field, including the fraction of the water column occupied by structure (*Moore, 2004*; *Hasegawa, Hori & Mukai, 2008*; *Luhar, Rominger & Nepf, 2008*), and the density and flexibility of structural elements (*Adhitya et al., 2014*; *Houser, Trimble & Morales, 2015*). Seagrass contributes tall, flexible structure, whereas oysters are stiff and typically shorter (although reefs can build up from gregarious settlement and particle trapping; *Walles et al., 2015*). Bivalves such as hard clams can intensify erosional processes and accordingly augment seston, even while drawing down phytoplankton biomass overall (*Porter, Mason & Sanford, 2013*). Similarly, benthic animals that bioturbate sediment or graze biofilms are associated with increased erodability of sediments (*Widdows et al., 2008*; *Guizien et al., 2014*). However, biogenic species might block the sediment surface and therefore reduce the surface area subject to resuspension. Dense seagrass in tidally dominated flow regimes is expected to slow laminar flow or divert flow away from sediment through skimming flow (*Koch & Gust, 1999*; *DeBoer , 2007*), whereas oyster reefs may increase turbulent flow with ancillary effects on particle transport (*Colden et al., 2016*).

On intertidal flats, the water transiting across a particular patch can vary dramatically in depth and motion through the tidal cycle. The consequences for water column seston are, however, difficult to predict because of covarying factors that may act in opposite directions. Specifically, shallow water near low tide may allow orbital motion of small waves to resuspend sediment, a process that no longer interacts with the bottom as depth increases (*Green, 2011*). Yet current speed could be increasing from slack low tide to mid-tide, increasing resuspension even as depth increases (*Widdows et al., 2008*; *Orvain et al., 2014*). At the same time, given evidence of vertical gradients of some particle types in shallow water (*Judge, Coen & Heck, 1993*; *Guizien et al., 2014*), benthic effects on water sampled at the surface may decline as depth increases simply due to distance from the bottom. As water depth increases, seagrass occupies a smaller fraction of the water column and its effects on surface water decline (*Koch, 2001*; *Luhar, Rominger & Nepf, 2008*). Similarly, deeper water depths dilute the effects of suspension-feeders on a per-volume basis. Thus surface water properties might be expected to become more homogeneous across a mosaic of habitat types as water depth increases; that is, depth × habitat interaction.

To empirically demonstrate the spatio-temporal heterogeneity of water properties on tidal flats, we sampled surface water over different habitat types, at multiple water depths experienced through the tidal cycle. We expected that water velocity would increase from low to mid-tide depths and that eelgrass would baffle currents. We expected water depth × habitat interactions for seston in which structured habitats influenced water properties at shallower depths, but heterogeneity was reduced among habitat types as surface water was farther from the sediment. For water properties of total suspended solids and chlorophyll-*a*, we evaluated both the static differences among habitat types based on point samples and the dynamic changes as water transited over bare tidal flat, eelgrass, or oysters.

## METHODS

### Study sites

The study took place at five low intertidal sites in Washington State, where eelgrass (*Zostera marina*), oysters (*Crassostrea gigas*), and unstructured bare mudflat co-occur near mean lower low water. The sites occupied distinct bays: Willapa Bay on the outer coast (46.5°N, 124.0°W), Samish Bay in north Puget Sound (48.6°N, 122.5°W), Case Inlet in south Puget Sound (47.3°N, 122.8°W), Port Gamble (47.8°N, 122.6°W) and Dabob Bay (47.8°N, 122.8°W) in Hood Canal, which is a natural fjord connected to Puget Sound (Fig. 1A). These tidal flats are exposed to air on extreme low tides and experience a mean tidal range of 2.1 m (Samish Bay, Willapa Bay) to 3.1 m (Case Inlet). Sediment type spans a range of sandy to muddy conditions (1–4% organic content, *Richardson et al., 2008*; *Yang et al., 2013*). The bays were selected to be representative of the diversity of eelgrass and oyster habitats in Washington State, so site was considered a random effect in all analyses. Recruitment of *C. gigas*, which is non-native, rarely occurs in Samish Bay and Case Inlet, so at these sites in particular, habitat patches of oysters were the product of aquaculture activities. Although these oysters can form vertical reefs, we focused on places where they occurred in clusters over large areas, consistent with the dimensions of other habitat patches. Each site was sampled over four days of spring tides in 2014 (Port Gamble 14–17 May, Case 26–29 May, Dabob 11–15 Jun, Willapa 25–28 Jun, Samish 10–14 Jul).

### Study design

The study design was nested, with measurements at three subsites in each of five sites. Each subsite consisted of patches of each of three habitats (bare, eelgrass, oysters), for a total of nine habitat patches per site. Thus, habitat type had three true replicates (i.e., distinct, interspersed) in each of the five sites. A patch had to exceed 20 m in its minimum dimension in order to be selected for study, and most were an order of magnitude larger than this. Each patch was sampled three times at different water levels, including upstream (initial) and downstream (final) paired samples. Overall, the design emphasized habitat type and water level as main effects. Across all five sites, we sampled 45 patches (evenly divided among bare, eelgrass, and oyster), with each patch sampled upstream and downstream at three water levels during the tidal cycle.

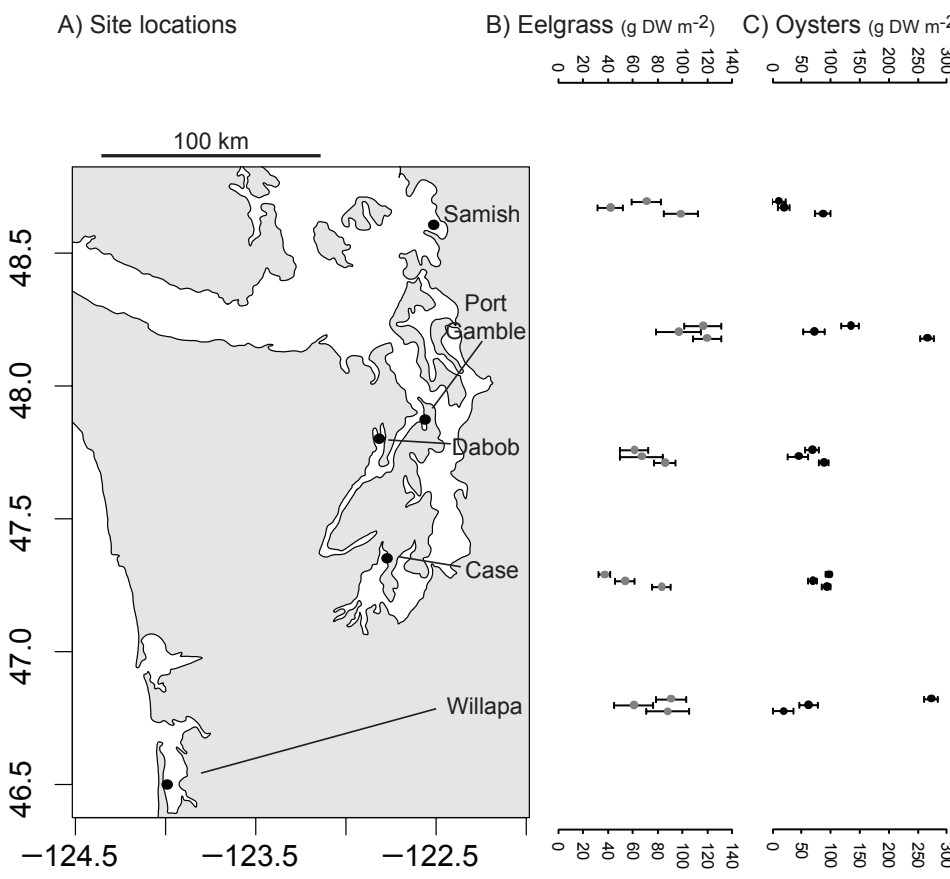

**Figure 1** **Biogenic species across study sites.** (A) Study sites in Washington State, USA. (B) Dry above-ground biomass of eelgrass (*Zostera marina*) in three habitat patches per site. (C) Estimated dry meat weight of oysters (*Crassostrea gigas*) from length-frequency distribution in three habitat patches per site. Error bars show SE of 10 quadrats per patch.

## Benthic sampling

Each of the 45 patches was surveyed at low tide, by placing 10 (0.25 m$^2$) quadrats at random intervals along a 50-m transect. In each quadrat in oyster habitat, we counted all live oysters and measured the first 10 for shell height (nearest 0.5 cm). In each quadrat in eelgrass, we counted eelgrass shoots and collected above-ground material of eelgrass. Eelgrass across these sites shows several ecotypic growth forms: smallest in Case Inlet (0.2 m average shoot length), intermediate in Port Gamble and Dabob Bay (0.3–0.6 m), and longest in Willapa Bay and Samish Bay (0.8–1.2 m). In quadrats in all habitats, we assessed shell cover and collected any macrophytes to assess biomass. Above-ground material of eelgrass and other macrophytes was dried separately (60 °C for 5 days) and weighed. Biomass of oysters was estimated from density and size-frequency based on the length-weight relationships of *Kobayashi et al. (1997)*, first at the quadrat level and then calculated per patch. Each of the 45 patches was described by the mean ($n = 10$ quadrats) and variation in above-ground biomass of primary producers and suspension-feeding oysters (Figs. 1B, 1C).

## Water sampling

To assess changes in water properties across patches, we tracked parcels of water with neutrally buoyant Lagrangian drifters consisting of a PVC frame at the surface of the water and a circular skirt, 80 cm in diameter and extending 10 cm into the water (Fig. 2, also used in *Wheat & Ruesink, 2013*). This approach enables sampling the same water over time, clarifying what was "upstream" and "downstream" at any particular time, and accounting for potentially variable initial conditions. We restricted our sampling to periods of light winds, since we observed that wind can decouple the drifter from strictly tracking the water. These light-wind conditions limited the development of surface waves, and water transport during sampling was driven by tidal currents. Eelgrass, oyster and unstructured tidal flat habitats were sampled concurrently. A waterproofed GPS (Garmin GEKO) was secured at the center of each drifter to ride on the water surface. Three drifts were carried out for each of the 45 habitat patches (27 drifts per site), spanning a range of water depths as depth changed during the tidal cycle. Water depth ranged from 0.1 m to ~2 m (94% of drifts occurred at water depths <2 m), which represented about half of the tidal amplitude during the sampling periods of spring tides. Most (85%) of drifts occurred on ebb tides. Drifts generally lasted half an hour, but duration was constrained by patch size and water velocity (Fig. 3). At the beginning of a drift, starting near a patch edge where water was flowing in to the patch, a person in a kayak collected water samples at ~0.3 m below the surface (or less in shallower water). Effort was made to collect water samples as close to the drifter as possible without disturbing its motion or the bottom. Water depth was evaluated by holding the paddle vertically and checking the water level at 0.1 m increments. As the drifter exited the patch, water samples were collected again. Each water sample collection consisted of three 300 ml Nalgene bottles for pigment analysis and one 1-L bottle for total suspended solids (TSS). All bottles were kept cold and dark and were filtered through 47 mm glass fiber filters (GF/F, 0.7 μm pore size) within three hours of collection. Samples for pigment analysis were placed in 10 ml of 90% (W/V) acetone and kept frozen and dark. At the end of the field season, these samples were measured for chlorophyll-*a* (Chl) via standard acidification procedure on a Turner Designs AU-10 fluorometer (*Welschmeyer, 1994*). Samples for TSS were filtered through pre-weighed filters, dried to determine mass of material, and combusted at 500 °C for three hours to determine organic content by loss-on-ignition. A few of these samples dried incompletely, generating large values for TSS (including some water) and for organic content (also including some water), and we censored nine values (eight drifts) with proportion organic <0 or >0.5. The time of collection of initial and final samples was recorded by the kayaker, which enabled later extraction of geo-positions from GPS units recording at 10 s intervals. The absolute distance between initial and final positions (in m, accounting for non-parallel longitude), divided by the duration of the drift, defined drift velocity.

## Data analysis

Five water properties were tested for depth and habitat differences. Two of these water properties were TSS and Chl collected at the initiation of each drift. Two were the change in TSS and Chl during each drift, using samples collected initially and finally in calculations of

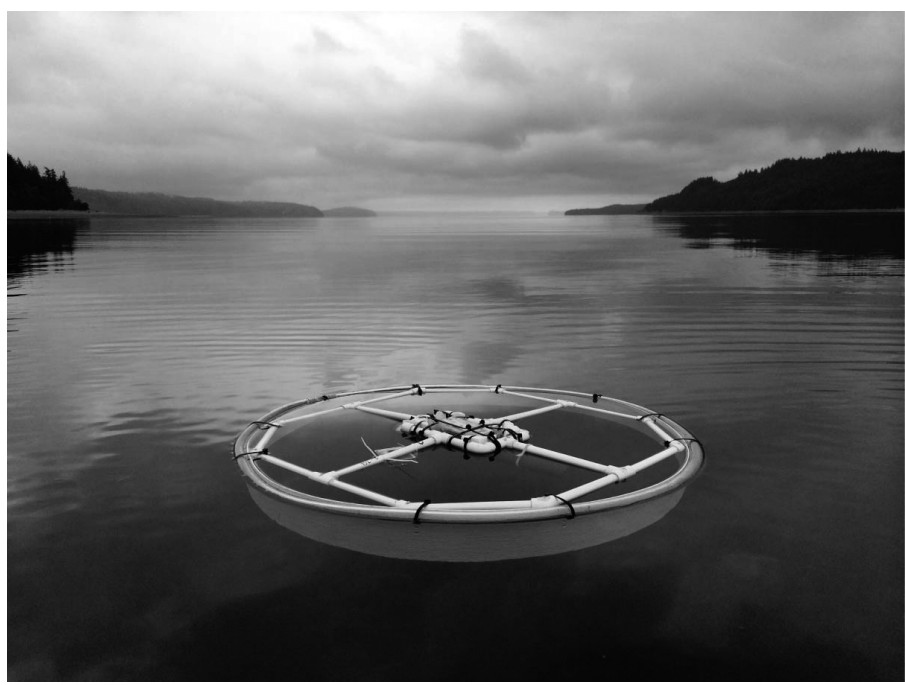

**Figure 2** **Neutrally buoyant Lagrangian drifter (diameter 0.8 m), with GPS unit at center.** Photo credit: Micah Horwith

$\ln(TSS_{final}/TSS_{initial})$ and $\ln(Chl_{final}/Chl_{initial})$. The fifth response variable was drift velocity. Each of these response variables was used in a linear mixed effects model, with fixed effects of habitat type, ln(water depth), and their interaction, and random effects of site and subsite in site. Three Chl samples collected simultaneously were averaged prior to analysis. The study design was unbalanced for any test of whether patterns differed between ebb and flood tides, or as a function of water residence time in a habitat patch. In exploratory analyses, these correlates explained little of the residual variation and were not considered further.

Quality of seston was evaluated through relationships of organic content and Chl to TSS. In these analyses, all samples from both initial and final collections for each drift were used. Response variables of ln(Chl) and ln(proportion organic) were used in linear mixed effects models, with fixed effects of habitat type, ln(TSS), and their interaction, and random effects of site and subsite in site. In all these analyses, any significant habitat effect was followed up by planned contrasts testing bare vs. eelgrass, and bare vs. oyster.

Residuals in all analyses were examined visually for normality, and TSS and Chl required ln-transformation. Following *Zuur et al. (2009)*, we tested for optimal random effects structure, which included random intercepts in all models. Statistical significance of predictors was set at $\alpha = 0.05$. However, when linear mixed effects models generate $P$-values near the border of significance, this provides weak evidence of their importance (*Zuur et al., 2009*). Linear mixed effects models were re-run with only significant factors in order to find coefficients for best-fit lines in visual display of data. Linear mixed effects analyses

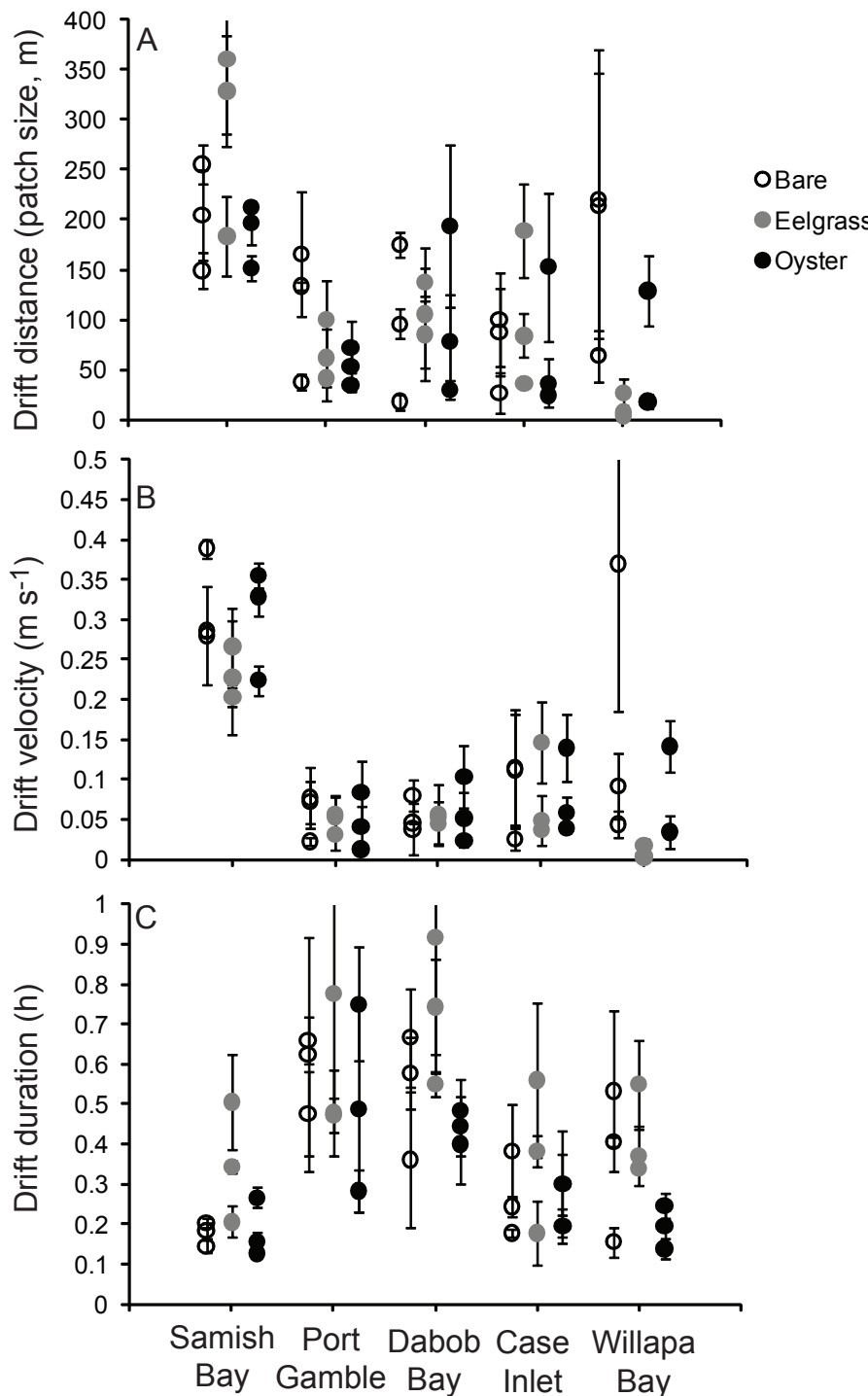

**Figure 3** **Characteristics of 135 drifts in shallow water across a mosaic of habitat types occupying intertidal flats in Washington State, USA.** Each point refers to one patch, with mean and standard errors calculated from the three drifts across that patch. (A) Distance from the start to end of each drift, demonstrating the dimension of each habitat patch. (B) Surface water velocity during drifts. (C) Duration of drifts.

**Table 1 Results of linear mixed effects models of water properties during each drift at five sites in Washington State, USA, in summer 2014.** Each column presents one of five water properties. Each row provides $F$ value ($P$ value) for main effects of habitat and depth, for their interaction, and for planned contrasts between biogenic habitats (eelgrass, oyster) and bare when habitat was significant. Random effects were site, and subsite in site. TSS, total suspended solids, mg $L^{-1}$. Chl, chlorophyll-a, µg $L^{-1}$. Samples were not used in analysis of TSS unless $0 <$ proportion organic $<0.5$. Some samples were not available for calculation of water velocity due to GPS malfunction.

|  | Response | | | | |
|---|---|---|---|---|---|
|  | $\ln(TSS_{initial})$ $n = 132$ | $\ln(Chl_{initial})$ $n = 136$ | Water velocity $n = 130$ | $\ln(TSS_{final}/TSS_{initial})$ $n = 127$ | $\ln(Chl_{final}/Chl_{initial})$ $n = 136$ |
| Habitat | 4.56 (0.013) | 14.9 (<0.0001) | 4.97 (0.009) | 0.02 (0.98) | 0.42 (0.66) |
| $\ln$(Depth) | 8.14 (0.005) | 3.61 (0.06) | 9.20 (0.003) | 7.58 (0.007) | 0.41 (0.52) |
| Habitat × $\ln$(Depth) | 0.56 (0.57) | 0.70 (0.50) | 0.88 (0.42) | 0.07 (0.93) | 0.17 (0.84) |
| Bare vs. Eelgrass | E>B $P = 0.003$ | E>B $P < 0.0001$ | E<B $P = 0.009$ |  |  |
| Bare vs. Oyster | O>B $P = 0.013$ | O>B $P = 0.019$ | O = B $P = 0.47$ |  |  |

were carried out with *nlme* (*Pinheiro et al., 2016*) in R (*R Core Team, 2015*). Characteristics of each of the 135 drifts have been archived (*Ruesink, 2018*).

## RESULTS

### Benthic composition of patches

Eelgrass patches contained 38–120 gDW m$^{-2}$ in above-ground biomass of eelgrass (Fig. 1B), with an overall moderate coefficient of variation (CV = SD/mean = 0.32, $n = 15$ patches). Oyster patches were more variable in estimated live oyster biomass (10–270 gDW m$^{-2}$, CV = 0.84, Fig. 1C) and ranged from 8 to 97% shell cover (mean 49%, CV = 0.53, $n = 15$, also positively correlated with live oyster biomass, $r = 0.52$). For both biogenic species, these densities are categorized as functionally dense, since fluid dynamics for sparse structure typifies cover <10% (*Bouma et al., 2007*). Other primary producers, in particular macroalgae, were present in some patches, averaging 2 gDW m$^{-2}$ in bare and eelgrass patches, but 20 gDW m$^{-2}$ in oyster patches, possibly reflecting the availability of hard surface for anchoring.

### Water properties at the initiation of drifts

Based on initial samples from each drift, the analysis of TSS revealed significant main effects of habitat and water depth, while Chl responded only to habitat type (Table 1). Higher concentrations of TSS and Chl were present in both biogenic habitats than over bare tidal flat, although the oyster-bare comparisons had $P$-values closer to $\alpha = 0.05$ that should be interpreted cautiously in linear mixed effects models. TSS declined as water depth increased, but Chl did not (Fig. 4, Table 1). The depth-related differences for TSS were in keeping with predictions of increased resuspension in shallow water, but the lack of depth × habitat interactions meant that the heterogeneity in surface water properties across habitat types was not muted as water level rose.

### Water properties during drifts

Based on initial and final samples from each drift, the analysis of $\ln(TSS_{final}/TSS_{initial})$ revealed a significant effect of water depth, but no factors were significant for

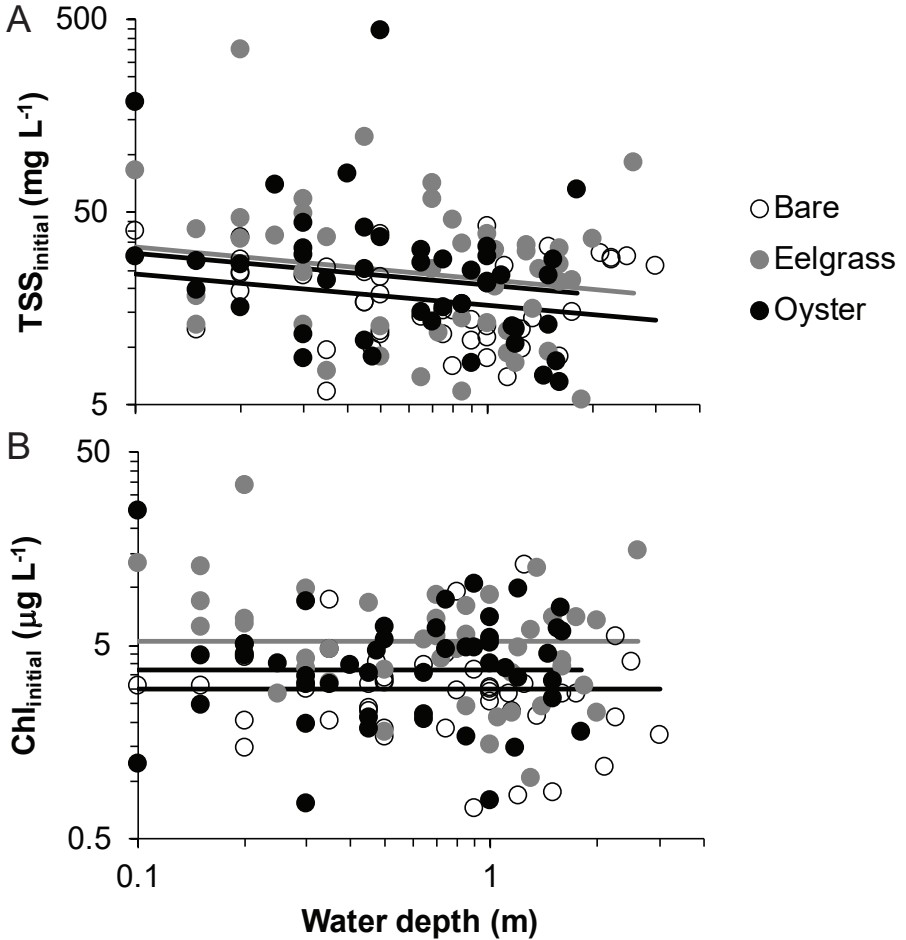

**Figure 4 Water properties at the initiation of each drift when water was at different depths over three intertidal habitat types at five sites in Washington State, USA.** (A) Total suspended solids. (B) Chlorophyll-*a*. Lines are plotted from the coefficients of linear mixed effects models including factors significant at $\alpha < 0.05$.

$\ln(\text{Chl}_{\text{final}}/\text{Chl}_{\text{initial}})$ (Table 1). Specifically, TSS continued to increase during drifts in shallow water but not when more water covered these patches (Fig. 5). No habitat differences or habitat $\times$ depth interactions emerged in analyses of changes in water properties during drifts (Table 1). Accordingly, distance to the sediment was important for the dynamics of TSS, but neither eelgrass nor oysters cleared the water of particles, regardless of water depth.

## Water velocity

Drift velocity showed main effects of habitat type and water depth but no interaction (Table 1). Dropping one fast outlier, velocity averaged 0.128 m s$^{-1}$ (SD 0.126, $n = 43$) across bare tidal flats, 0.115 m s$^{-1}$ (SD 0.114, $n = 43$) over oysters, and 0.085 m s$^{-1}$ (SD 0.096, $n = 44$) over eelgrass (Fig. 3). Drifts over eelgrass were 33% slower relative to bare tidal flats, but this effect did not depend on water depth. Drifts increased in velocity when more water

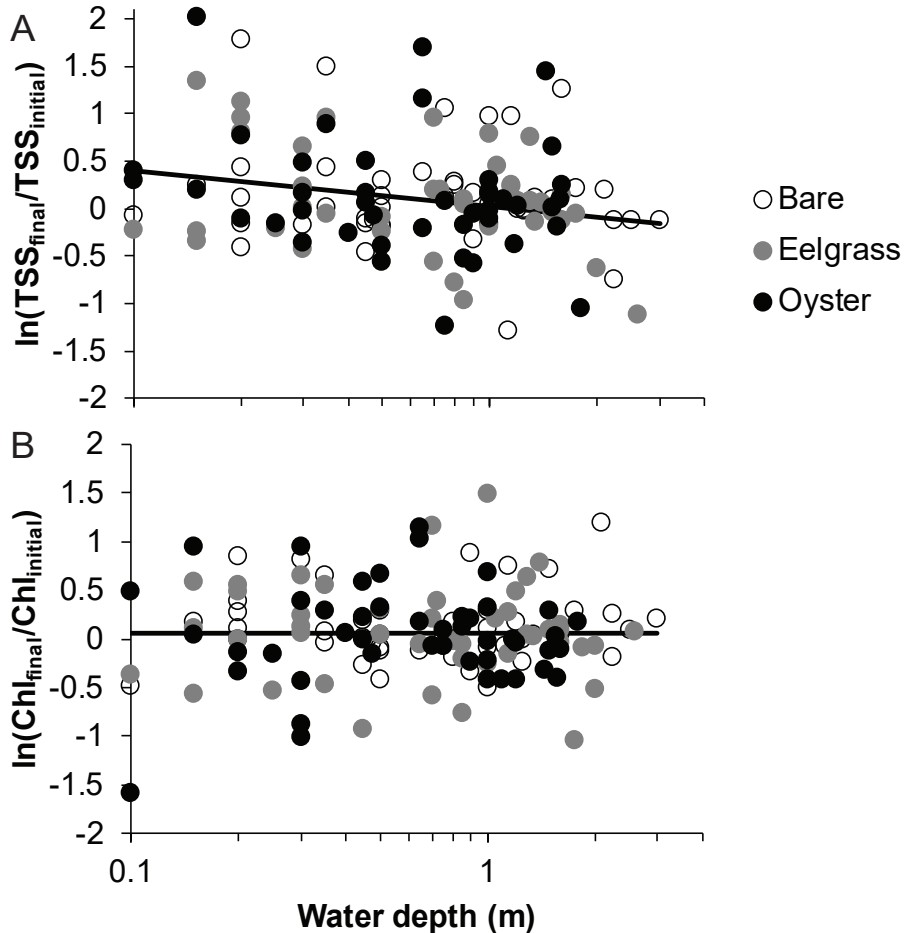

**Figure 5** Change in water properties during drifts when water was at different depths over three inter-tidal habitat types at five sites in Washington State, USA. (A) Change in total suspended solids, based on the log-ratio. (B) Change in chlorophyll-*a*, based on the log-ratio. Lines are plotted from the coefficients of linear mixed effects models including factors significant at $\alpha < 0.05$; habitat was not significant in these models, so a single relationship is shown.

was over habitat patches, consistent with currents through the tidal cycle (Fig. 6). Across all drifts, 62% were slower than $0.1$ m s$^{-1}$, and the 49 that were faster than this included all 27 drifts in Samish Bay (Fig. 3).

## Seston quality

Proportion organic was significantly related to TSS, whereas Chl additionally differed by habitat type. Chl was positively related to TSS, rising from 4 to 6 µg/L across a range of TSS that varied by two orders of magnitude (5 to 500 mg/L; Fig. 7, Table 2). Across this same range of TSS, organic content declined, shifting from 30% to 10% (Fig. 7). Seston characteristics differed by habitat for Chl, which was higher for a given amount of TSS in eelgrass relative to bare tidal flat (Table 2).

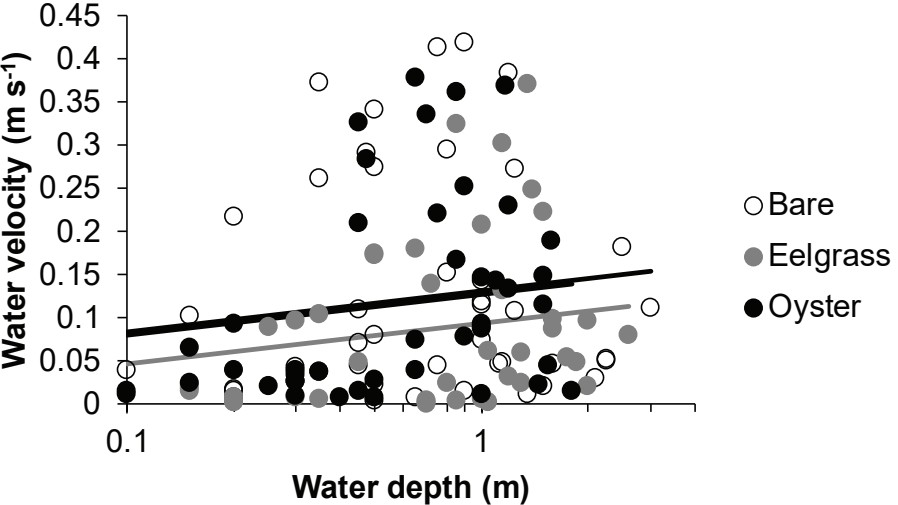

**Figure 6** **Water velocity during drifts when water was at different depths over three intertidal habitat types at five sites in Washington State, USA.** Lines are plotted from the coefficients of a linear mixed effects model including factors significant at $\alpha < 0.05$. Although the residuals in this plot appear to violate assumptions of normal distributions, the distributions were suitable for analysis in the linear mixed effects model, which accounted for site and subsite as random effects.

## DISCUSSION

In our measurements of material carried by surface water across tidal flats, TSS was inversely related to water depth and tended to increase further during transit in shallow water (Figs. 4 and 5). Such increases in particle loads in water flowing across tidal flats have been documented previously in the field (*Guizien et al., 2014*), and resuspension often governs shallow water properties, in terms of both the amount and characteristics of seston (*Gacia & Duarte, 2001*; *Newell & Koch, 2004*). Resuspension increases with current speed (up to a point) in flume studies, while the water velocity able to lift particles is contingent on bottom properties (*Widdows et al., 2008*; Ovain et al., 2014). Thus, TSS in our surface water samples was affected more by vertical water motion (depth or distance to sediment) than by horizontal water motion (current speed), which acted in an opposite direction to observed results. That is, as more water covered the tidal flat, it traveled faster (Fig. 6) and carried less TSS at the surface (Figs. 4 and 5). Contrary to expectations, no water depth × habitat interactions were apparent in TSS and Chl. Regardless of depth, highest TSS and Chl concentrations were observed over eelgrass, where reduced water velocity was expected to lead to particle settling (Fig. 4). We found little evidence of top-down control by oysters, given that change in Chl over oysters was similar to that in other habitat types (Fig. 5).

Both biogenic species acted as seston-mobilizers based on comparisons of initial drift values (Fig. 4), but this effect was not apparent for changes during drifts (Fig. 5, Table 1). Specifically, in initial samples, surface water over eelgrass and oysters contained twice the concentration of TSS relative to bare: 38.0 (7.9 SE), 36.9 (9.8 SE), and 19.3 (1.5 SE) mg L$^{-1}$, respectively. This habitat-specificity in initial values was also apparent in Chl: 6.4 (0.8 SE),

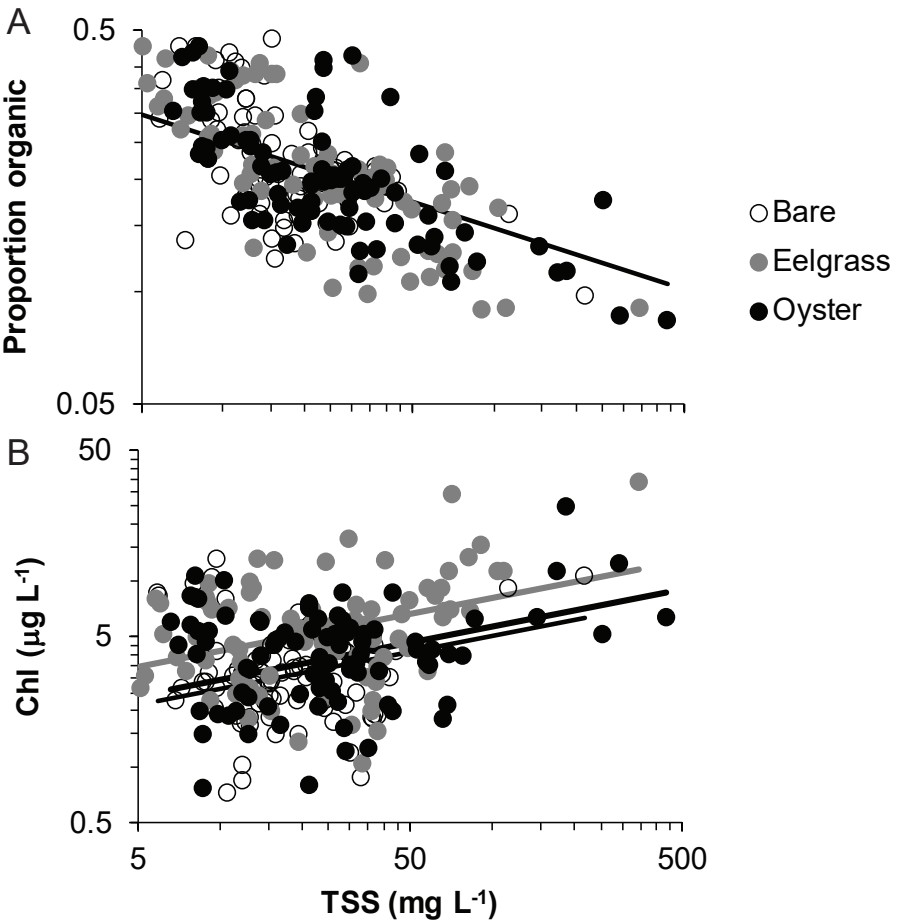

**Figure 7** **Quality of seston in shallow water during drifts over three intertidal habitat types at five sites in Washington State, USA.** (A) Organic content, and (B) Chlorophyll-*a* concentration in water varying in total suspended solids. Samples include both initial and final measurements for each drift. Lines are plotted from coefficients of linear mixed effects models including factors significant at $\alpha < 0.05$. Accordingly, a single relationship is shown for proportion organic, where habitat was not significant, but separate lines for chlorophyll-*a* due to water over eelgrass containing higher concentrations than over bare.

4.7 (0.6 SE), and 3.7 (0.3 SE) µg L$^{-1}$. The finding that biogenic habitats did not further influence seston differentially with respect to bare suggests that the mosaic of intertidal water properties is established at patch borders, for instance because erosional processes tend to be enhanced at leading edges of structured habitats (*Adams et al., 2016*). Factors expected to result in build-up of effects during drifts, such as suspension feeder activity, may not differ among habitats as expected from oysters alone, given that we did not sample for infaunal suspension-feeders (i.e., *Peterson & Black, 1987*).

Despite the absence of predicted water depth × habitat effects, differential responses to water depth for surface TSS and Chl can help infer causes of spatio-temporal heterogeneity over these tidal flats. TSS was inversely related to depth in terms of both concentration and change during drifts, but the amount and dynamics of Chl were unrelated to water depth (Figs. 4 and 5, Table 1). Evidently, resuspension of material on these tidal flats

**Table 2  Results of linear mixed effects models of water properties using initial and final samples for each drift at five sites in Washington State, USA, in summer 2014.** Each column contains a distinct water property: chlorophyll-a (Chl), and proportion organic. Each row provides $F$ value ($P$ value) for main effects of habitat and total suspended solids (TSS), for their interaction, and for planned contrasts between biogenic habitats (eelgrass, oyster) and bare when habitat was significant. Random effects were site, and subsite in site. Samples were not used in analysis unless 0 < proportion organic < 0.5. The interaction effect (habitat $\times$ ln(TSS)) was close to $P = 0.05$ in this mixed effects model, therefore interpreted cautiously.

| | Response | |
| --- | --- | --- |
| | ln(Chl) $n = 261$ | ln(proportion organic) $n = 262$ |
| Habitat | 24.2 (<0.0001) | 0.10 (0.90) |
| ln(TSS) | 57.3 (<0.0001) | 146.1 (<0.0001) |
| Habitat $\times$ ln(TSS) | 3.16 (0.044) | 0.49 (0.61) |
| Bare vs. Eelgrass | E > B $P < 0.0001$ | |
| Bare vs. Oyster | O = B $P = 0.03$ | |

predominantly mobilized non-photosynthetic material, thus augmenting TSS without changing Chl. This explanation is supported by the negative relationship between proportion organic and TSS in analyses of seston quality (Fig. 7). Although Chl was positively related to TSS, the slope of this relationship was small, given ca. 50% increase in Chl over a range of TSS spanning two orders of magnitude (Fig. 7), and insufficient to drive identical depth-related patterns for particulates as a whole and for Chl. In other studies, benthic microalgae were found to be lifted into the water column under conditions of lower water energy than can mobilize mineral particles (*Orvain et al., 2014*) and to distribute without much gradient in the water column (*Guizien et al., 2014*). Due to evidence in our study that primarily inorganic particles were resuspended in shallow water, we interpret this to mean that benthic microalgae were already mobilized under the full range of drift velocities and depths, or that populations of benthic microalgae were not well developed on these tidal flats.

In retrospect, a large gap in our study is that we do not have coupled measurements of sediment properties for all drift locations. It is reasonable to expect that eelgrass and oysters could influence sediment properties, and that a source of seston could be biodeposits in the case of oysters and locally-produced organic matter in eelgrass. However, in some past work we have found no consistent differences in the sediment of these habitat types relative to bare tidal flat (*Richardson et al., 2008*). In the present case, we cannot test how much of the variation in seston among drifts is a function of the sediment type over which the water passes, therefore precluding an assessment of this potential mechanism underlying the heterogeneity in shallow water properties that we documented.

Attenuation of water movement by seagrass has been observed in many field and lab studies and is largely dependent on the current speed (*Fonseca et al., 1982*), driver of water motion (e.g., wind vs. tide; *Koch & Gust, 1999*), habitat configuration (patch vs. continuous; *Worcester, 1995*), and seagrass density and shoot length (*Moore, 2004*; *Hasegawa, Hori & Mukai, 2008*). In our study, water velocity was reduced over eelgrass relative to other habitat types (Fig. 6, Table 1). Thus, all combinations of canopy height and water depth

represented conditions suitable for modifying surface flow. It may be necessary to include drifts at still deeper water depths associated with extreme high tides to see evidence of any habitat × depth interaction, which should emerge because flow reduction extends only to a factor of two of canopy height (*Luhar, Rominger & Nepf, 2008*).

Why, then, did this slower flow not lead to deposition of particles and a clearing of the water column in eelgrass? Instead, we speculate that eelgrass on these tidal flats may increase water turbulence and/or surface area hosting fragile microalgal epibionts. Seagrass biomass tends to be concentrated off-bottom, and the near-bottom material is gathered into leaf sheaths. Accordingly, reduced drag may enable faster near-bottom flow that fosters erosional rather than depositional processes (*Madsen et al., 2001*; *Koch et al., 2006*). Key tests in flumes have been carried out on small, dense morphotypes of eelgrass (e.g., 1,000 shoots m$^{-2}$ of ∼20 cm length; *Fonseca & Koehl, 2006*), whereas most eelgrass morphotypes found in Washington State are larger, sparser, and therefore more likely to result in resuspension. The second possibility is contribution to seston from the microalgae and trapped sediments on the surface area of the eelgrass leaves. This layer of fine fuzz can represent almost as much dry mass as the eelgrass itself during summer months (*Nelson & Waaland, 1997*; *Ruesink, 2016*). Mobilization of epiphytes into the water column could underlie the higher quality (as Chl) of seston in eelgrass relative to other habitats in our study (Fig. 7). Spcifically, 50 mg L$^{-1}$ of TSS is associated with about 5 µg L$^{-1}$ of Chl (Fig. 7), but 50 mg of material scraped from eelgrass leaves contains 100 µg of Chl, yet a moderate organic content (13%, *Ruesink, 2018*). Overall, suspended materials in water moving through eelgrass may depend on morphologically-mediated differences in bottom turbulence resuspending benthic particles or picking up materials from the large surface area of eelgrass leaves.

One other issue regarding water velocity is worth noting here, which is the rapid flow in Samish Bay (Fig. 3), where the bathymetry has a shallow grade. Eelgrass in Washington State exhibits two spatial configurations: flats (areas with extensive broad shallows such as river deltas and pocket beaches) and fringes (areas with linear eelgrass distribution due to steep bathymetry; *Berry et al., 2003*). Samish Bay was our only site where sampling overlapped eelgrass flats. There, the bathymetry was conducive to both extensive eelgrass and rapid flow, resulting in drift durations similar to other sites. Overall, the flat vs. fringe dichotomy needs further examination for the engineering of water properties by biogenic species. In this study, with one "flat" and four "fringes", it was necessary to consider site a random effect.

Many studies of water properties directly over shellfish beds have documented measurable depletion of water column resources (*Grizzle et al., 2006*; *Grizzle, Greene & Coen, 2008*; *Grangere et al., 2010*; *Plutchak et al., 2010*; *Wheat & Ruesink, 2013*). Downstream concentrations of Chl are typically lower than upstream concentrations, but not as different as would be expected from scaling up filtration rates measured in the laboratory (*Grizzle, Greene & Coen, 2008*; *Wheat & Ruesink, 2013*). Here we found little evidence of oyster filtration reducing Chl or particle loads above patches (Table 1). This weak effect may be due to overall low oyster biomass, that is, averaging <100 gDW m$^{-2}$ relative to 400 gDW m$^{-2}$ where drawdown has been documented in Washington

State (*Wheat & Ruesink, 2013*). Also, our baseline condition of "bare" tidal flats contained unknown numbers of infaunal suspension-feeders. Because seston did not decline during drifts in any habitat type, our data point towards resuspension as a factor that may complicate assessments of the filtration capacity of shellfish. Other researchers have noted rapid variation in processing speed in response to food quality and quantity (*Barillé et al., 1997*) and filtration failing to scale with abundance due to the collective consequence of reef structure (*Colden et al., 2016*). The accumulation of empirical evidence regarding filtration as an ecosystem service is essential to understanding the conditions under which bivalves can improve water quality, and by how much.

## CONCLUSIONS

Ecological interest in resuspension processes in shallow water has been motivated by understanding food web linkages and subsidies among habitats. Sediment resuspension has a number of potential ecological feedbacks to the biogenic species considered in this study. Resuspension provides a potential benefit to suspension-feeders as they may use benthic microalgae or other organic particles in their diet (*Kang et al., 2003*; *Herman et al., 2000*; *Van Oevelen et al., 2006*). In contrast, for seagrass, resuspension contributes to light attenuation and provides a mechanism whereby alternative stable states can occur, when seagrass clears its own water (*Ralph et al., 2007*; *DeBoer , 2007*), which was not the case in our study. Our empirical examination of seston in surface water passing across tidal flats revealed spatial heterogeneity that mapped on to habitat mosaics and shifted during the tidal cycle. Potential food resources were elevated in eelgrass, as has been demonstrated in other species of seagrass as well (*Judge, Coen & Heck, 1993*; *Lebreton et al., 2011*). We worked at particularly low water levels in a dynamic tidal environment, which may help explain why resuspension appeared as a primary driver of seston. The habitat-specific mosaic that we documented may be a feature of summer sampling, given seasonally high biomass of eelgrass, although also providing a best-case temperature scenario for oysters to feed (*Ren & Ross, 2001*). The relative rates of deposition, filtration, and resuspension deserve further scrutiny across sediment types, more extreme water depths, and morphologies and epiphyte loads of eelgrass, which could be additional factors involved in the spatial transfer of resources in coastal environments.

## ACKNOWLEDGEMENTS

We appreciate access to field sites from Taylor Shellfish and Port Gamble Sk'lallam Tribe. Michael Hannam, Collin Gross, Joy Polston-Barnes, and Dolores Sare contributed to field work.

### Funding

Funding was provided by the Washington Department of Natural Resources through interagency agreement with the University of Washington (IAA 14-164). The funders had no role in study design, data collection and analysis, decision to publish, or preparation of the manuscript.

### Grant Disclosures

The following grant information was disclosed by the authors:
Washington Department of Natural Resources.

### Competing Interests

Cinde R. Donoghue and Micah J. Horwith are employed by the Washington Department of Natural Resources which provided funding for this project.

### Author Contributions

- Jennifer L. Ruesink conceived and designed the experiments, performed the experiments, analyzed the data, prepared figures and/or tables, authored or reviewed drafts of the paper, approved the final draft.
- Cinde R. Donoghue and Micah J. Horwith conceived and designed the experiments, performed the experiments, contributed reagents/materials/analysis tools, authored or reviewed drafts of the paper, approved the final draft.
- Alexander T. Lowe performed the experiments, authored or reviewed drafts of the paper, approved the final draft.
- Alan C. Trimble conceived and designed the experiments, performed the experiments, authored or reviewed drafts of the paper, approved the final draft.

### Data Availability

Ruesink J (2018), "Data from: Comparison of shallow water seston among biogenic habitats (eelgrass, oysters, bare) on tidal flats", Mendeley Data, v2 http://dx.doi.org/10.17632/9y8xrhvhmy.1.

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
