# Peer review of "Comparison of shallow-water seston among biogenic habitats on tidal flats"

_PeerJ, doi:10.7717/peerj.6616_

## Round 0.1 · original submission · Major Revisions

I agree with reviewer 1 that there is still some (but not much) work to be done before your article can be accepted for publishing. When addressing the reviewers questions please be sure to answer all the questions and integrate the requested parts especially from reviewer 1.

Reviewer 1 ·

Basic reporting

Overall, the manuscript was well written. A few areas, however, need to be rewritten to improve clarity and facilitate reader comprehension. Jargon throughout should be kept to a minimum, and definitions provided whenever terms are introduced that may not be universally known or agreed upon.

L36-39 - Please rephrase to improve clarity.

L45 – Please define foundation species.

L162 – “[…] within three hours.” – Of collection?

L310-311 – Please rephrase to improve clarity.

Figure 7a - Is the y-axis a log scale? Unclear given just two values, while most of the other figures have had log-scaled y axes.

Experimental design

The experimental design was largely adequate for the data. I do have a couple of questions that I would like the authors to address.

1) Why did you subset the data rather than use contrast statements? L188

2) Including tidal stage (ebb vs. flood) may help explain some of the variation in the data. Incoming tides are more directly interacting with the benthos than ebb tides. While water depth may partially address this, mid flood at 1-m depth has much more recently been close to the benthos than mid ebb at 1 m.

3) Including patch size as a random effect may also help identify habitat specific differences, and may be especially useful to identify changes in final/initial TSS and Chl, if they exist.

Validity of the findings

The data are robust and statistically sound. Pending how the authors address the questions about experimental design, some of the following comments may be moot.

1) You have made the assumption that bare tidal flats will be less effective at mitigating overlying TSS and Chl. Were these barren seascapes, that may be true, but the authors should also consider that tidal flats can support a large biomass of filter feeders (e.g., see https://doi.org/10.4319/lo.1987.32.1.0143), which may be as capable of removing seston as the oysters in the study, especially given the moderate biomasses recorded in the surveys. A few benthic grabs, both for sediment characteristics as noted by the authors, as well as infauna would have helped to address this.

2) In lines 270-272, you discuss how eel grass had higher initial TSS and Chl than the other habitats. Figure 4 demonstrates this, but the log-scaled y-axis doesn't make the assessment clear. I recommend including a bar plot (or equivalent) without a log-scaled y-axis to make this clearer.

3) The findings should be clearly stated throughout to apply to the surface waters in your study areas. Given that all sampling occurred within the top 0.3 m of the water column, and no measures of vertical mixing were taken, it is important to note that there may have been changes to the seston as it passed over the habitat patches, but that they weren't evident in the surface waters.

4) I'm not sure that I buy the argument in lines 320-322 that the higher Chl over eelgrass patches was because of mobilization of epiphytes. Most epiphytes have evolved to attach to their host, and in the low-wind conditions of your study, I'm not convinced that this is the best explanation. Please provide other support from the literature to support this speculation. The elevated Chl could also be more of the background water conditions at your sites. The oysters and infauna of bare patches can more directly interact with the overlying water column than the filter feeders below the eelgrass. There may be a slight decoupling of the filter feeders in eelgrass from the water column as local hydrodynamics within the overhead eelgrass dominate.

Additional comments

Overall, this is a good article with good data. There are some issues that need to be addressed, including the statistical analyses and subsequent interpretation. I'm looking forward to seeing the revisions, as I believe that the study was overall well-conceived and worthy of eventual publication.

·

Basic reporting

The reporting is clear, unambiguous, well-structured, and complete.

Experimental design

The research question is well defined, the methods are rigorous and well tested, and the analysis is well executed. Data are archived and openly available.

Validity of the findings

The conclusions are well supported by the results of the current and previous studies.

Additional comments

This is a brilliant, well designed and well executed study that advances our understanding of the role habitat-formers play in structuring their physical habitat properties and the habitats around them. The one and only [minor] question that comes to mind is where is the source of material? If oysters and eelgrass are seston-mobilizers, how is that material being deposited on these habitats in the first place? It may be worth speculating on this somewhere in the discussion. Maybe on windy days, which are difficult to sample with Lagrangian drifters? That goes against my expectation though.

Reviewer 3 ·

Basic reporting

The language of this article was professional and clear except a few minor comments:

Line 24: Instead of "During drifts" should this be "Across the eelgrass habitat..." Using "during drifts" implies (to me) that the drift changed the results and not the habitat.

Line 36-39: The article chose to put a list in parentheses here. I believe these are important for the article, especially the particulates, I would recommend taking this information out of parentheses.

Line 267-270 and 276-278: Both of these paragraphs start with "The conclusions...". I found this to be distracting from the flow and also the sentences. Especially 267-270, I had to re-read this sentence a couple of times to understand the meaning.

The background and context was quite clear to me. I'm not sure that we will find the genus change to Magallana from Crassostrea to be long-standing.

The figures and tables are clear and the raw data is shared. I might remove Figure 5 as the results are negative and explained within the text.

Experimental design

The article meets the PeerJ standards. I found the research questions to be well-defined and filling a meaningful gap in our understanding of the PNW oyster-eelgrass ecosystems. This MS represents a rigorous investigation performed to a high technical and ethical standard. The methods were described in sufficient detail to replicate.

Validity of the findings

I can see that the study took sufficient effort, but I am wondering if the authors considered performing the same drifts over multiple seasons? If so, would you expect the results to be the same? It might be worth adding the potential effects of seasonality to the discussion.

---

## Round 0.2 · accepted · Accept

Thank you for this nice review of your article. We are now able to accept it for publication.

Reviewer 1 ·

Basic reporting

no comment

Experimental design

no comment

Validity of the findings

no comment

Additional comments

The authors have done an excellent job of addressing my previous questions and comments, and I have no further questions or concerns. The article is well written and easy to comprehend. I look forward to seeing it in publication.

Reviewer 3 ·

Basic reporting

No comment

Experimental design

No comment

Validity of the findings

No comment

Additional comments

Thank you for your clarifications!